# Investigating the structural changes due to adenosine methylation of the Kaposi's sarcoma-associated herpes virus ORF50 transcript

**Konstantin Röder**[1]*, **Amy M. Barker**[2], **Adrian Whitehouse**[2], **Samuela Pasquali**[3]*

**1** Yusuf Hamied Department of Chemistry, University of Cambridge, Cambridge, United Kingdom, **2** School of Molecular and Cellular Biology and Astbury Centre of Structural Biology, University of Leeds, Leeds, United Kingdom, **3** Laboratoire CiTCoM, UMR 8038 CNRS, and Laboratoire BFA, UMR 8251 CNRS, Université de Paris, Paris, France

\* kr366@cam.ac.uk (KR); samuela.pasquali@u-paris.fr (SP)

## Abstract

Kaposi's sarcoma-associated herpes virus (KSHV) is a human oncovirus. KSHV relies on manipulating the host cell N6-methyl adenosine (m6A) RNA modification pathway to enhance virus replication. Methylation within a RNA stem loop of the open reading frame 50 (ORF50) increases transcript stability via the recruitment of the m6A reader, SND1. In this contribution we explore the energy landscapes of the unmethylated and methylated RNA stem loops of ORF50 to investigate the effect of methylation on the structure of the stem loop. We observe a significant shift upon methylation between an open and closed configuration of the top of the stem loop. In the unmethylated stem loop the closed configuration is much lower in energy, and, as a result, exhibits higher occupancy.

## Author summary

In this article we present the investigation of the change in structure of an RNA regulatory molecule upon a change in the chemistry of one of its bases. Eukaryotic RNAs contain more than 100 different types of chemical modifications, which can fine-tune the structure and function of RNA. Since RNA systems need to adopt a specific 3D shape to be functional, it is important to understand how a chemical modification impacts the structure adopted. Using the computational technique of energy landscape explorations, that is exploring what structures are available to the system at a given energy, we are able to characterise the RNA before and after the modification, and understand what the main differences between the ensembles of structures, which can be adopted by the system, are.

In this work, we present our results of this investigation on an oncogenic virus-encoded RNA. We show how a chemical modification at a precise location of the native structure affects the system globally, inducing a rearrangement of parts of the structure, which are far away from the modification site.

**Data Availability Statement:** The databases containing all minima and transition states and the analysis scripts used are available online (DOI:10.

5281/zenodo.5647374). All software used is publically available.

**Funding:** KR is funded through a Fellowship from the Cambridge Philosophical Society, and was awarded a visiting fellowship by the Universite de Paris. AMB and AW are funded by the MRC (MR/R010145/1). AW has funding from the BBSRC (BB/T00021X/1, BB/V006363/1). The funders had no role in study design, data collection and analysis, decision to publish, or preparation of the manuscript.

**Competing interests:** The authors have declared that no competing interests exist.

## Introduction

Kaposi's sarcoma-associated herpes virus (KSHV) is a human oncovirus associated with Kaposi's sarcoma, a highly vascular tumor of endothelial lymphatic origin, and several other AIDS-associated malignancies [1]. Like all herpesviruses, KSHV has a biphasic life cycle consisting of latent persistence and a lytic replication phase. Notably, both phases are required for KSHV-mediated tumorigenesis. Expression of a single KSHV-encoded protein, the replication and transcription activator (RTA) protein [2], is necessary and sufficient for the transition between latency and lytic replication, leading to the activation of the complete lytic cascade resulting in infectious virions.

RTA is encoded from the open reading frame 50 (ORF50)—and its expression is stimulated by various cellular cues such as plasma cell differentiation [3] and hypoxia [4]. RTA activates transcription of lytic genes by directly interacting with RTA-responsive elements (RREs) found in multiple lytic gene promoters, or indirectly via interactions with cellular transcription factors, particularly RBP-Jκ, AP-1, and Oct-1 [2].

N6-methyladenosine (m6A) is the most prevalent internal modification of eukaryotic messenger RNAs (mRNAs). Due to recent transcriptome-wide m6A mapping of multiple viruses, it is becoming evident that there is an interplay between m6A-decorated viral RNA and the host cellular m6A machinery, resulting in the modulation of viral replication output [5]. Several studies have demonstrated that the KSHV transcriptome is heavily m6A methylated [6]. The m6A sites are targets for protein recognition by so-called m6A reader proteins, with different proteins leading to a variety of biological fates, from promotion of RNA degradation [7, 8] to enhancement of translation [9]. In addition to direct recognition, indirect recognition is also possible via the so called m6A switch mechanism. Here, the methylation modification has been shown to destabilise and alter RNA structures allowing the recruitment of protein binding partners [10, 11].

In KSHV, m6A modification of the ORF50 RNA transcript leads to recruitment of the m6A reader Staphylococcal nuclease domain-containing protein 1 (SND1) [6]. SND1 binding to the ORF50 RNA stabilises the transcript, resulting in effective lytic replication. SND1 recruitment to ORF50 RNA is m6A dependent as binding is impaired upon depletion of the m6A methyltransferase, METTL3. Furthermore, depletion of SND1 results in destabilised RNA, lower levels of RTA and impaired lytic replication [6]. m6A modification of the ORF50 transcript occurs at a classical DRACH (D = A, G or U; R = A or G; and H = A, C or U), GGACU, motif situated in a 43mer hairpin. The hairpin is relatively unstable, due to a large number of unpaired nucleotides and resulting weak base pairing, and further destabilisation, likely associated with structural changes of the stem loop, is necessary for SND1 binding [6].

While it is likely that the m6A modification alters the RNA structure and as a result its binding affinity to SND1, it is not clear what structural change occurs, and how it allows the recruitment of a protein binding partner. An additional factor complicating our understanding of this process is the fact that only two-dimensional structural models are available for the RNA stem loop. Hence, simulations are valuable not only in identifying the changes between the methylated and unmethylated system, but furthermore in describing the structural ensemble in general. As the structural heterogeneity displayed by RNA stem loops and the associated slow dynamics complicate experimental and computational studies, an enhanced sampling approach may yield additional insight into the structural and kinetic properties of the stem loop, and the effect of the methylation. This information may be obtained by an explicit exploration of the energy landscape of the unmethylated and methylated stem loop. While this approach has not previously been applied to study epigenetic changes, it has been successfully used to study mutations in a different stem loop [12], as well as other non-canonical nucleic

acid structures, and is known to produce insight into structural ensembles and the transitions between them [13, 14].

Here, we present the free energy landscapes for the unmodified and the m6A modified stem loop within the ORF50 transcript. We observe two main structural ensembles, differentiated by the orientation of A22, the m6A modification site. In one ensemble, the nucleotide is inside the stem loop, not accessible to potential binding partners (*in*-configuration), while in the other ensemble, it is pointing away from the stem loop (*out*-configuration). While the unmodified RNA shows a strong preference for the *in*-configuration, the m6A modification allows for a higher population of the *out*-configuration, through structural alterations based on changes in the intricate pattern of stabilising interactions. We further observe a key change in the lower part of the stem loop, where a large bulge can also stabilise or destabilise the stem loop. Both processes are interconnected, and hence required for binding, as observed in experiment [6].

## Materials and methods

The energy landscapes of the unmodified and methylated stem loops were explored using the computational potential energy landscape framework [13, 14]. Below we give a brief summary of the simulation techniques in this work; a detailed guide to exploring energy landscape in this way can be found elsewhere [15]. In addition, S1 Text section A contains further detail, including discussions of certain choices within the modelling process.

The two-dimensional structures predicted by Baquero-Perez *et al.* were used as starting points with three leading U nucleotides at the 5′ end. This leading UUU motif means the full size of the simulated stem loop is 44 nucleotides, but we start the labelling at 0 to be consistent with previous work. From these two-dimensional structures, initial three-dimensional configurations were obtained from RNAComposer [16] (see S1 Text Fig A). RNAComposer allows the translation of the two-dimensional into a three-dimensional structure via fragment assembly. This structure exhibits the correct canonical interactions, but will not necessarily be the optimal structure with respect to non-canonical interactions and flexible regions, such as bulges. Therefore, basin-hopping (BH) global optimisation [17–19] was used to obtain low energy structures using physical modelling, optimising these remaining interactions. We conducted three sets of five BH runs. The first set used the unmodified structure from RNAcomposer as input. For the second set, we used the same structure but applied the m6A methylation. These two sets of basin-hopping runs were used to seed the energy landscape explorations. The final set was run for the unmodified set, but starting from an unfolded structure. This final set probed partially folded structures, but as the energy difference between them and the low energy folded structures was very large (around 80 to 100 kcal/mol), these structures are unlikely to be significant and we did not repeat this set of runs for the modified molecule. Each of the runs within the sets consisted of 150,000 BH steps, with grouprotation moves [20, 21] to create new configurations, and a convergence criterion for minima of $10^{-6}$ kcal mol$^{-1}$ Å$^{-1}$. The 100 lowest energy minima from each run where used to seed the databases to explore the energy landscapes by locating discrete paths.

Discrete pathsampling [22, 23] was employed to create a kinetic transition network [24, 25]. Transition state candidates were obtained using the doubly-nudged elastic band (DNEB) algorithm [26–28], and the actual transition states were located via hybrid eigenvector-following [29]. We obtain the free energies using the superposition approach with a harmonic approximation [30]. The rate constants are calculated using the new graph transformation (NGT) algorithm from the kinetic transition network [30, 31]. We represent the energy landscapes as disconnectivity graphs [32, 33], which faithfully represent the topography of the

energy landscape (i.e. the folding funnels and their substructure) and the energy scale. In these graphs, a structure is associated to a point on the vertical axis according to its free energy. A vertical line is drawn upwards from each point. Lines are merged when the energy of all transition states connecting the structures is exceeded, which is drawn in a discrete manner to allow proper visualisation. The resulting horizontal ordering is such that structures with easier transitions are closest, and therefore faithfully shows the funnel structure of the energy landscape.

Throughout, the ff99 [34] force field with the Barcelona $\alpha/\gamma$ backbone modification [35] and the $\chi$ modification for RNA [36, 37] was used with implicit solvent ($igb$ = 2 [38]). The parameters for the modified nucleotide were taken from the standard AMBER library [39].

Structures in the energy landscape are assigned to funnels and define structural ensembles. Low-energy structures in each ensemble are analysed using Barnaba [40] to detect base pairs, stacking and to compute all torsions in the backbone, sugar and pucker. For each structure, the overall number of canonical, non-canonical and stacking interactions is recorded for each nucleotide. All these values are then averaged to give the average local behaviour of the ensemble.

## Results

The key results of this study are the free energy landscapes for the unmodified and the 22m6A-modified RNA stem loops. Disconnectivity graphs [32, 33] for these two systems at 310 K are shown in Figs 1 and 2, respectively. The three distinct structural ensembles associated with funnels are called $A$, $B$ and $C$ in the unmodified system, and the corresponding structural ensembles in the m6A-modified system are $A^*$, $B^*$ and $C^*$.

In both landscapes, we see a broad separation of the structures into configurations where A22 is pointing inwards (*in*-configuration) and the nucleobase interacts with the surrounding nucleotides, and configurations where A22 is pointing away from the stem loop (*out*-configuration).

A significant difference is the relative energies of these structural ensembles. In both cases, the *in*-configuration is observed at the bottom of the free energy landscape. The *out*-configurations in the unmodified stem loop are between 22 and 30 kcal/mol higher in free energy. In contrast to this large gap, the methylation significantly lowers this energy gap to around 10 kcal/mol. In addition, multiple alternative stem configurations exist for the *in*-configuration, leading to substructure with some distinct subfunnels. For the unmodified system, these subfunnels are roughly 7 to 10 kcal/mol higher in free energy than the global minimum, while the m6A modification of A22 leads to more distinct substructure at slightly higher energies around 9 to 12 kcal/mol. A full discussion of the important structural features is given later. It should be noted here that the energy of the local minimum for the structure obtained originally from RNAComposer is about 70 kcal mol$^{-1}$ higher than the global minima for the unmodified and the modified system. There are two reasons for this observation. Firstly, the structure is in an *out*-configuration, and therefore already significantly higher in energy than the lowest energy minima in the $A$ and $A^*$ ensembles. Secondly, while the canonical base pairing is correct, due to the constraints from the two-dimensional structure, the non-canonical interactions and the flexible regions are not optimally arranged.

As a consequence of the change in relative energies of the structural ensembles, we predict a significant change in the transition rates between the different structures. All predicted rate constants and the associated equilibrium constants are given in Table 1. The transition from the *in*- to the *out*-configuration for A22 in the unmodified system is 10 orders of magnitude slower than the reverse process ($1.535 \cdot 10^{-10}$ s$^{-1}$ vs. $1.427$ s$^{-1}$). Thus, we do not expect any significant population of the *out*-state at biologically relevant temperatures.

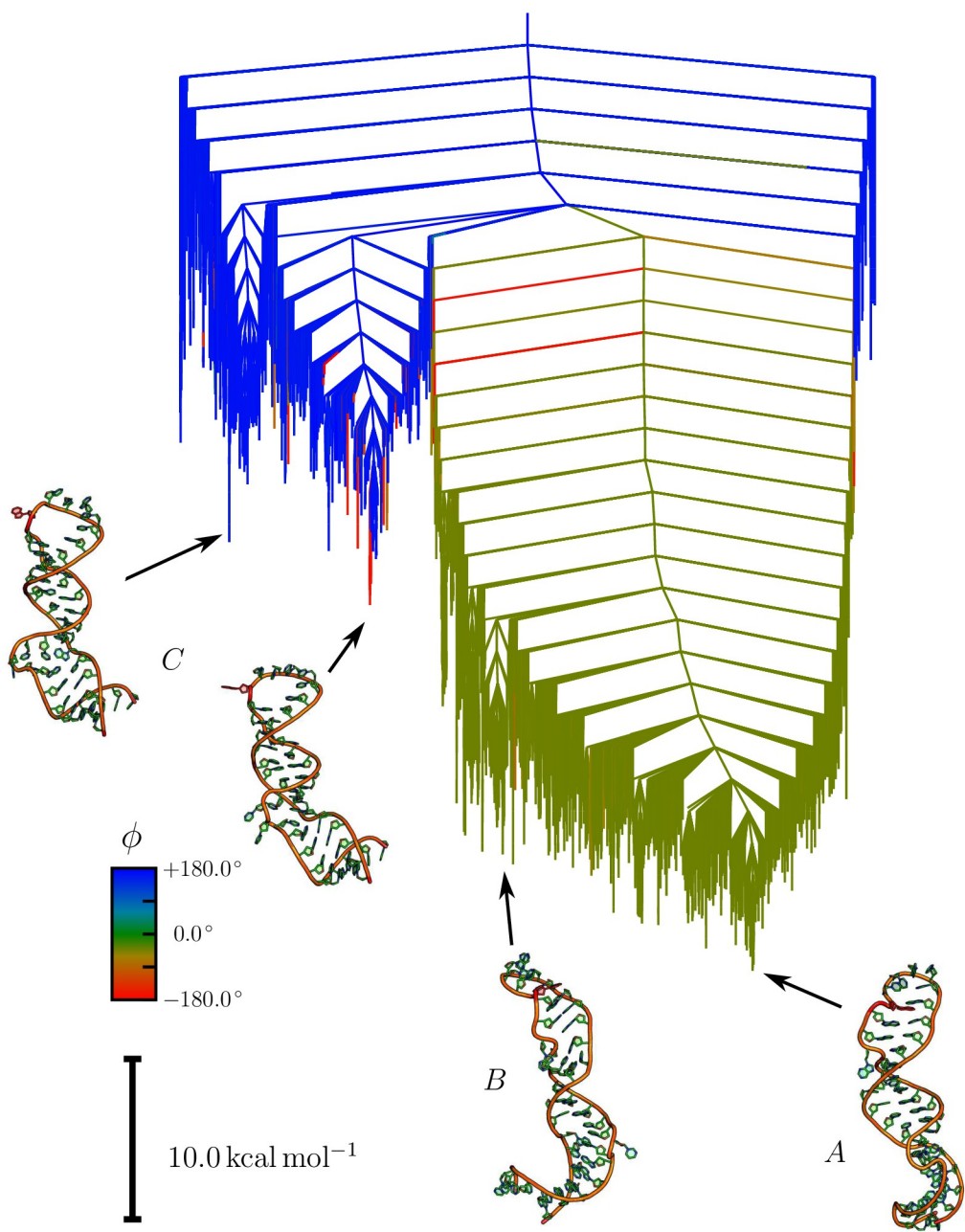

**Fig 1. Free energy disconnectivity graph for the native loop.** The free energy disconnectivity graph at 310 K is shown for the unmethylated RNA stem loop. Representative structures are shown for each funnel, in which A22 is highlighted in red. Two main funnels are observed, one where A22 is orientated inside the loop (*A* and *B*) and another where A22 is pointing outwards (*C*). The former set of structures is much lower in free energy. The outwards orientation of A22 at the top of the stem loop is visible in the example structures shown on the graph for *C*, in contrast to the closed arrangement in *A*. The colouring scheme highlights the *in/out*-configuration of A22, and is based on $\phi$, which is the dihedral angle formed by the basepair below A22, and the nucleotide itself, where values around 0 indicate an *in* configuration (green), and values close to ±180˚ are *out*-configurations (blue and red). Some structural variation is observed in both major funnels, resulting in the emergence of smaller subfunnels. These variations are mainly located in the lower parts of the stem loop, leading to different stem configurations (see for example *A* and *B*).

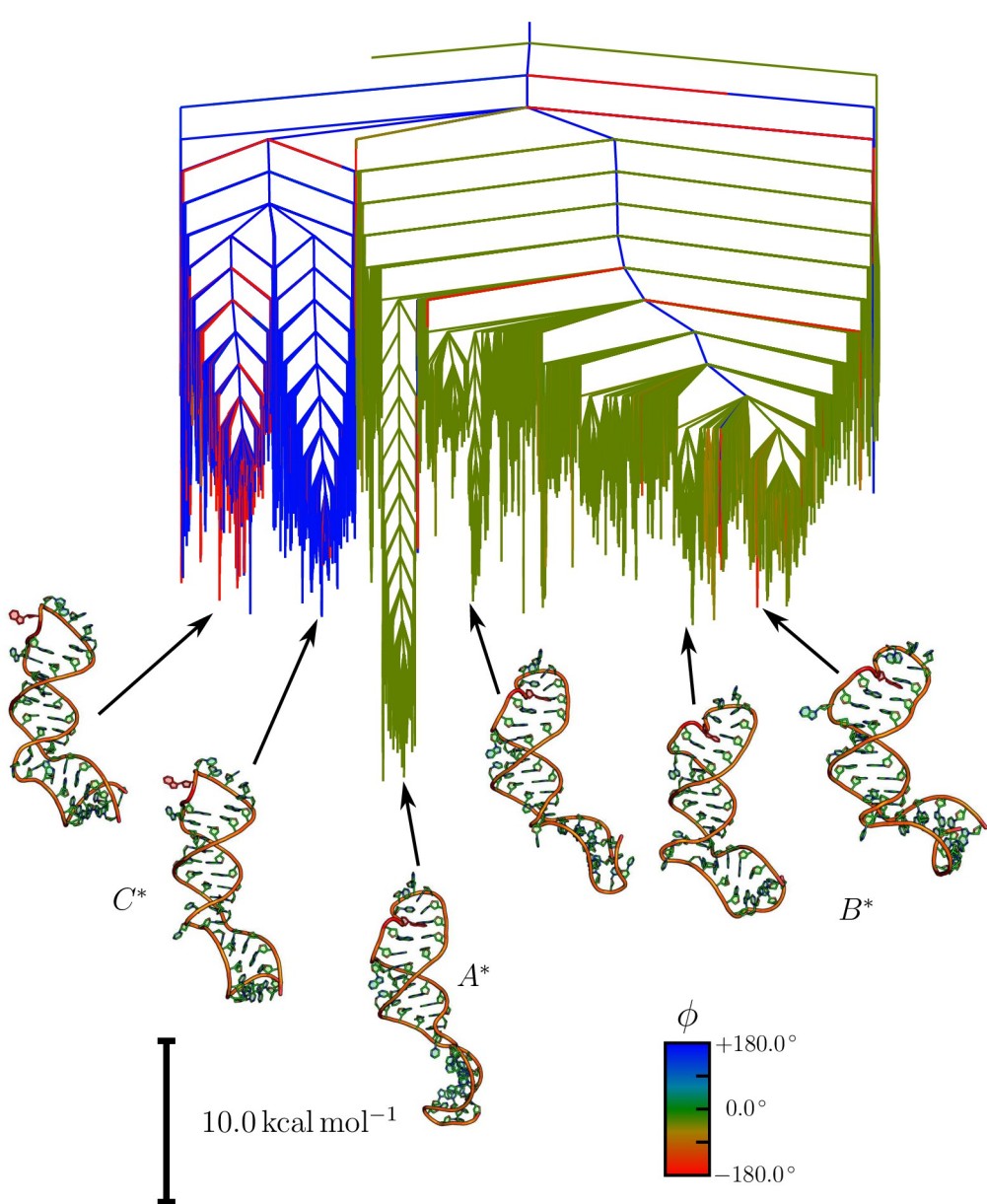

**Fig 2. Free energy disconnectivity graph for the modified stem loop.** The free energy disconnectivity graph at 310 K for the methylated RNA stem loop exhibits significant changes compared to the unmethylated stem loop (see Fig 1). Representative structures are shown for each funnel, in which m6A22 is highlighted in red. The structural ensembles are preserved, but their relative energies are significantly altered. Firstly, the free energy of the *out*-configuration ($C^*$) is lower, and in many cases comparable to the *in*-configurations. The set of lowest free energy structures is still an *in*-configuration ($A^*$), but a large number of *in*-configurations ($B^*$) are similar in energy to the *out*-configurations ($C^*$), mainly due to changes in the lower part of the RNA stem loop.

The picture for the m6A-modified stem loop is different, but also complicated by the topography of the energy landscape. For the unmethylated system, the substructure in the main funnel only consists of small, shallow subfunnels. In the modified system, the subfunnels containing structures with the *in*-configuration are more clearly separated. This separation arises from alterations in the lower stem loop. When considering a transition from $A^*$ to $C^*$, the changes in the lower and upper stem loop therefore need to be treated as two distinct, but

**Table 1. Rate constants and equilibrium constants between _A_, _B_ and _C_ at 310 K.**

| System | Set of minima | $k_{out \leftarrow in}$ (s$^{-1}$) | $k_{in \leftarrow out}$ (s$^{-1}$) | $K_{eq}$ |
|--------|---------------|-----------|-----------|--------|
| A22 | $A + B \leftrightarrow C$ | $1.535 \cdot 10^{-10}$ | 1.427 | $9.30 \cdot 10^{9}$ |
| m6A22 | $A^* \leftrightarrow C^*$ | $4.588 \cdot 10^{-15}$ | $2.943 \cdot 10^{-8}$ | $6.41 \cdot 10^{6}$ |
| m6A22 | $B^* \leftrightarrow C^*$ | $2.262 \cdot 10^{-8}$ | $2.943 \cdot 10^{-8}$ | 1.30 |

Rate constants and the related equilibrium constants at 310 K for the unmodified (A22) and the modified (m6A22) system. For the methylation, the two different transitions are given, as described in the full text.

connected events. Considering a lowest minimum to lowest minimum transition ($A^*$ to $C^*$ in the disconnectivity graph), we still observe a significant bias towards the _in_-configuration, although the equilibrium constant is three orders of magnitude smaller ($6.41 \cdot 10^6$ compared to $9.30 \cdot 10^9$). If we consider a transition including the higher energy minima for the _in_-configuration ($B^*$) the forward and backward rates are nearly identical and we compute an equilibrium constant around unity. A key change in both cases is a significant slow down in the reaction rate from $C^*$ to $A^*$ and $B^*$ by eight orders of magnitude compared to $C$ to $A$. This result means that the change in relative energies will lead to a significant population of the _out_-configuration in the A22-N6-methylated system.

## Heat capacity curves show structural transitions

The changes in the rate and equilibrium constants will significantly impact the population of the different structural ensembles. A useful way to illustrate these changes, and further link them to an energy scale, is to consider the heat capacity curves for both systems. Peaks in heat capacity curves are associated with phase transitions between different states, and in the case of molecular ensembles may be interpreted as the transitions between different structural ensembles. Furthermore, the heat capacity is directly linked to the occupation probabilities of different configurations, and each peak in the curve can therefore be analysed in terms of increasing and decreasing occupation probabilities [41]. This analysis allows to identify two sets of structures, each one dominant on opposite sides of the phase transition (peak). As we go further from the peak on one side, the structure dominant on the other side become less and less likely to be observed, while at the peak the two structures co-exist with equal probability.

In Fig 3, the heat capacity curve for the unmodified stem loop is shown, alongside representative structures for the transitions obtained by the analysis introduced above. In these schemes, the structures on the left hand side of the arrows are the structures dominant below the peak, while the right hand side structure are the ones observed above. As the calculation of the heat capacity curve requires some approximations, and furthermore our simulations use implicit solvent, the curves should only be interpreted qualitatively, with the most important information coming from the difference between the two curves for the two systems. Some additional commentary is provided in the S1 Text, Section D.

Two important transitions are observed, labelled P1 and P2 in Fig 3. P1 is the lowest energy feature, which is associated with a rearrangement in the lower stem region. This transition corresponds to higher occupancy of the higher energy structures in the main funnel on the landscape. The second feature, P2, is the transition from _in_ to _out_.

We observe the same two transitions for the m6A-modified stem loop, as shown in Fig 4, also labelled as P1 and P2. Importantly we observe a distinct shift to lower energies for both transitions. We can associate the energy scale for the peaks with Boltzmann population proportions, and find that the P2 peak in the modified system is at energies accessible to around

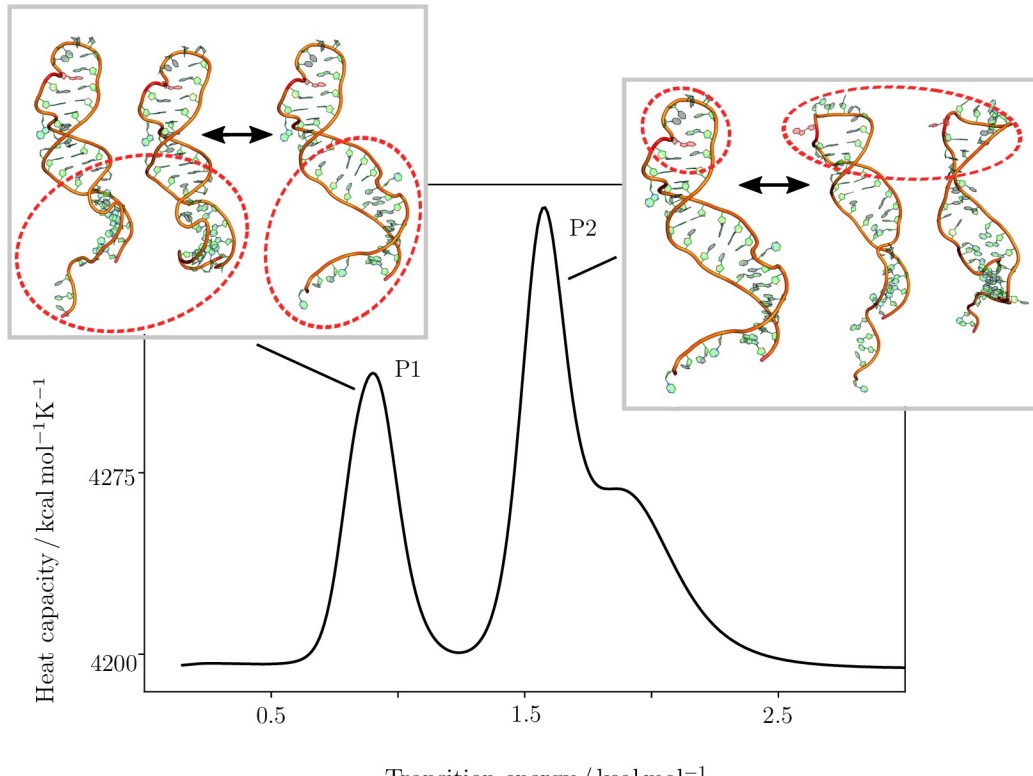

**Fig 3. Heat capacity curve for the unmodified RNA stem loop.** Heat capacity curve for the unmethylated RNA stem loop shows three distinct features. Analysing the contributions from local minima to the associated transitions [41] reveals a medium temperature transition between different configurations of the lower stem loop configuration (P1), and a high temperature transition between in- and out-configurations for A22 (P2). The shown structures are representative of the structural ensembles on either side of the transition. Where more than one structure is shown, this choice was made to provide a fair representation of the structural variation observed. For more detail on the shoulder observed for P2, see the S1 Text, section D and Fig D.

26% of molecules, while in the unmodified system this percentage shrinks to only 7%. This shift is the key observation, as the transition associated with P2 is likely required for binding, and is significantly more likely in the modified system.

## Structural variations observed in the energy landscapes

Having established that the m6A-modified stem loop shows significant alterations in the occupation of the structural ensembles on the energy landscape, we will need to look at the structural ensembles more closely to identify the key changes introduced by the N6-methylation of A22, and how these changes alter the transition mechanism from the *in-* to the *out-*configuration.

The details of the local behaviour in the four structural ensembles A, C, A* and C* are given in Fig 5 where we can observe the variations in base pairing (canonical and non-canonical), stacking and pucker configuration for each nucleotide (details of the 3D structures are reported in S1 Text Fig B). The first global observation is that the canonical base pairing is well preserved across the four ensembles, with only one or two variations and with A exhibiting the most canonical pairs. Based on these canonical base-pairings we can identify secondary structural elements common to all ensembles. A first helical region is where pairing between nucleotides G4 to A7 with nucleotides U39 to U42 is observed (H1), with an additional base pair between U3 and A43 often occurring. A second helical region (H2) is more variable, but its

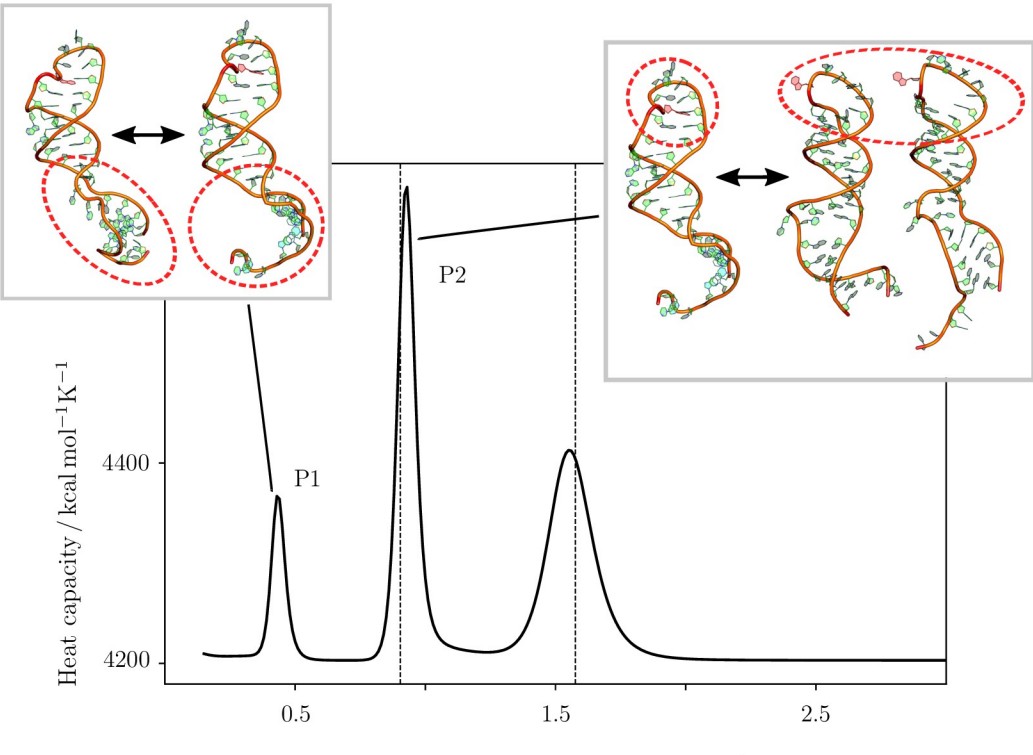

**Fig 4. Heat capacity curve for the A22-methylated RNA stem loop.** Heat capacity curve for the A22-methylated RNA stem loop shows three distinct features similar to the unmodified stem loop (see Fig 3), but the peaks are shifted to lower temperatures. For reference, the peak positions in the unmodified loop are shown as dashed lines. The transition between different configurations of the lower stem loop configuration (P1) is now at temperatures well below room temperature, and the transition between in- and out-configurations for A22 (P2) at medium temperatures. The shown structures are representative of the structural ensembles on either side of the transition. Where more than one structure is shown, this choice was made to provide a fair representation of the structural variation observed. For more detail on the third peak at higher transition energies, see S1 Text, section D and Fig D.

core contains nucleotides G13 to C17 paired with G32 to C37, where either C33 or U34 are not paired. Two other key regions are the apical loop (L) sitting above the m6A-modification site containing nucleotides C23 to U27, and the bulge in the lower stem (B), which is formed by A8 to C11.

The overall puckering is also well preserved across the ensembles with only a few, but key, transitions from C3'-endo to C2'-endo in going from A to C and from A to A* and C*. These changes involve nucleotides that significantly rearrange from one ensemble to another by either switching base-pairing partner or swinging outward or inward with respect to the helical stems. Non-canonical pairing and stacking exhibit more variability across the ensembles with changes spread all across the structure. We can observe a loss of non-canonical pairing in going from A to C as well as in going from A* to C*, and to a lesser extent going from A to A*. The parameter showing the largest variability is stacking with many significant changes occurring in all parts of the molecule, including the two helical stems and the apical loop.

From all these observations we detail below the most significant structural changes across the four ensembles.

**The *in*-configuration in the unmodified and m6A-modified stem loop.**   A comparison of the secondary structure of the unmodified and modified stem loop in their lowest energy

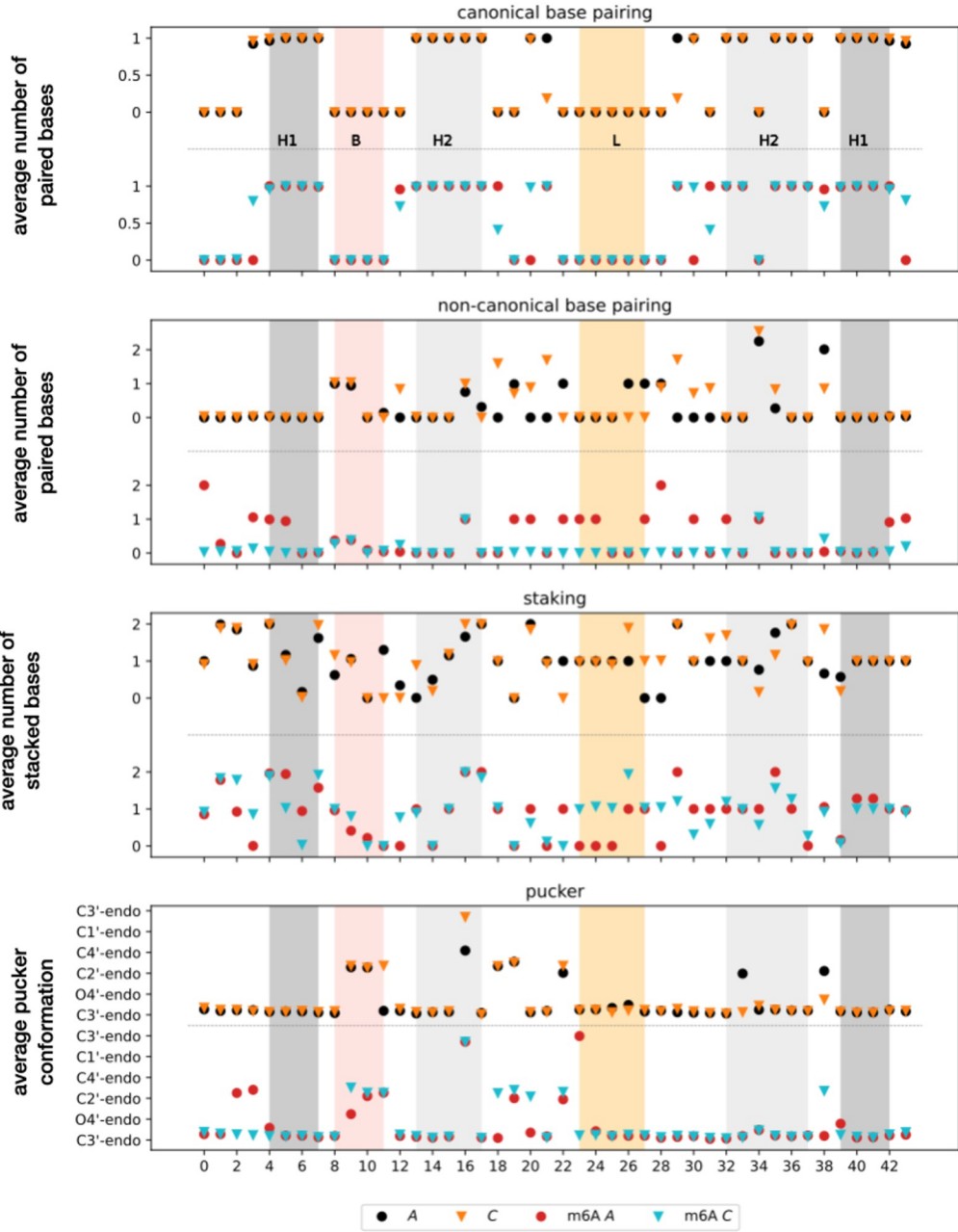

**Fig 5. Local structural properties.** For each of the four low-energy ensembles, A, C, m6A A (A*) and m6A C (C*), we report the behaviour of each nucleotide in terms of number of canonical and non-canonical base pairing, independently of the partners, number of stacking interactions formed, and puckering conformation, as extracted from Barnaba software. Reported numbers are the average over the ensemble of structures in the corresponding funnel, computed for each nucleotide. Each panel is composed of an upper part for the native unmethylated system (A and C) and a lower part for the methylated system (A* and C*), separated by a horizontal dashed line. For puckering we express the values in terms of the classification in conformations instead of angular values (for this reason C3' corresponding to an angle of 0 or 360 is repeated). Shaded regions represent the different secondary structure elements with helices in grey (H1 and H2), bulge (B) in pink, and apical loop (L) in yellow.

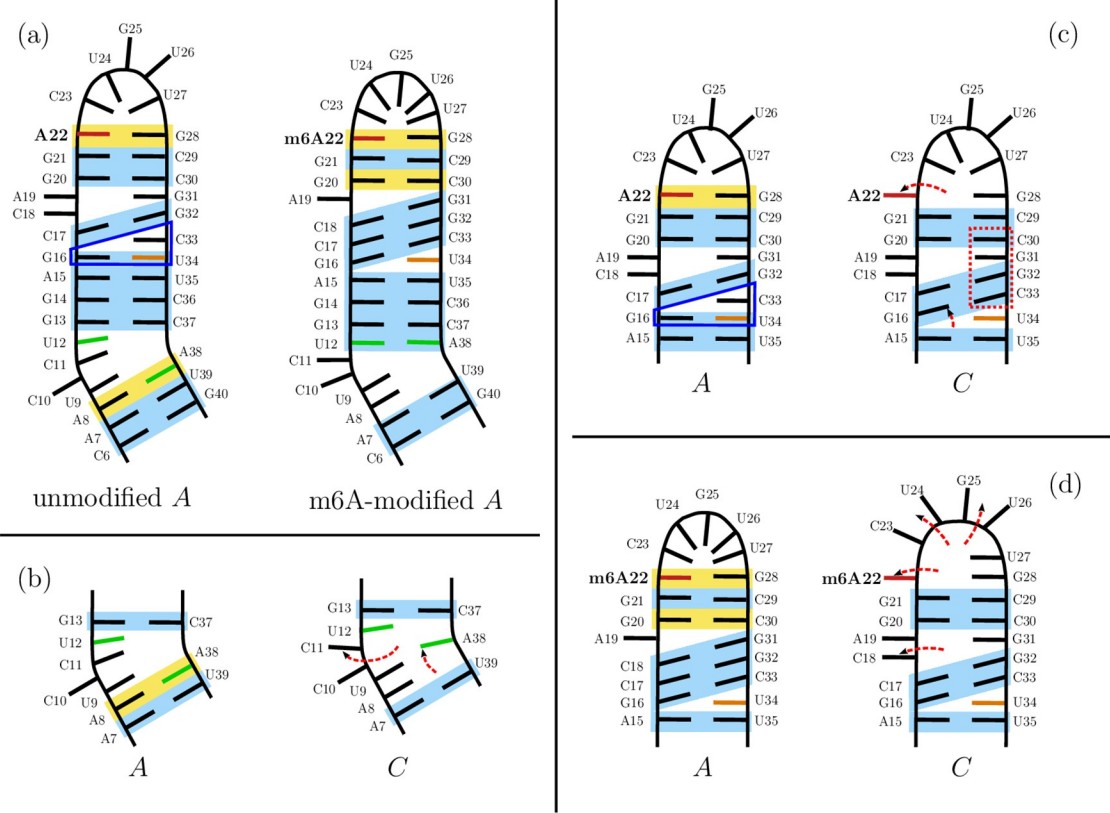

**Fig 6. Two-dimensional structures for the key structural ensembles.** Schematic two-dimensional structures are shown for the key structural ensembles, and their key features are highlighted and compared. Canonical base pairing is indicated in blue, and non-canonical base pairing interactions in yellow. A22 is highlighted in red, and other important residues are also highlighted, namely U34 (orange), U12 and A38 (both green). Key variations between structures are indicated with red dashed arrows, indicating the change in nucleobase orientation. The triplet formed by interaction between C33 and the G16-U34 basepair is indicated by a blue frame and important stacking in a red frame. The data used in this figure is derived from the ensemble properties, which are shown in Fig 5 in more detail. (a) Comparison of the lowest energy ensembles for the unmodified and m6A-modified stem loop. (b) Changes observed in the lower bulge for the *in*-configuration (*A*) and the *out*-configuration (*C*) of the unmodified system. (c) Changes in the upper stem loop between the *in*-configuration (*A*) and the *out*-configuration (*C*) for the unmodified system. (d) Changes in the upper stem loop going from the *in*-configuration (*A\**) to the *out*-configuration (*C\**) for the m6A-modified stem loop.

ensembles (*A* and *A\**) is shown in Fig 6(a), highlighting some important changes. The first set of changes are located in the upper stem loop in helix H2. A triplet formed between the paired G16-U34 and C33 in the unmodified stem loop is altered in the modified stem loop, as the G16-U34 base pair is replaced by a G16-C33 canonical base pairing. Moreover, this change is accompanied by the base pairing of C18-G31 in the modified molecule. In addition, more non-canonical interactions in the apical loop L are observed for the methylated system, leading to a tightly packed loop, where C23 to U27 are all pointing inwards. Last but not least, the bulge B undergoes significant rearrangement, with U12 and A38 canonically paired in the modified system, replacing the non-canonical A8-A38 pairing in the unmodified system. As a result, the methylated molecule has C11 pointing out alongside C10. The helix H1 is less affected, and the only change is the existence of the additional non-canonical base pairing between A8 and A38 in the unmodified system on top of the helical stack.

The changes between them might therefore be summarised as follows. The helical region H2 is significantly extended in the m6A-modified loop. The stacking in the modified stem loop is altered compared to the unmodified loop (see Fig 5). This modification is a result of the alteration of A22, such that m6A22-G28 is in a different configuration, with m6A22 sitting

somewhat further outside the stacked nucleotides, likely due to the additional space requirement by the methyl group. This alteration allows stacking for G28 and m6A22, with alignment of more nucleotides along H2, leading to a larger stacked region including more paired bases, albeit at the cost of more strain in the backbone. This extension of the helical region impacts the lower stem, in particular the bulge B, and leads to C11 pointing outwards. This change in B is very similar to the change in this region observed in the *out*-configuration for the unmodified stem loop (see Fig 6B).

**Changing from *in*- to *out*-configuration in the unmodified stem loop.**   The difference between the *in*- and *out*-configuration in the unmodified stem loop is characterised by changes in two main regions: the bulge B in the lower stem loop illustrated in Fig 6B, and changes in helix H2 and the apical loop L as shown in Fig 6C. The changes in B are C11 pointing outwards, and the non-canonical interaction between A8 and A38 being lost. As a result, we observe kinking between H1 and H2, and associated changes in the backbone. This change in B and at the lower end of H2 is also observed in higher energy structures with the *in*-configuration for A22, and can be followed by rearrangements at the top of H2, which are the loss of the G16-U34 base pair and the formation of the G16-C33 base pair instead. This alternative pairing leads to stacking from this new base pair up to the apical loop on top of C33. The stacking stabilises this configuration in which A22 can swing out. While this leads to a loss of contacts in the apical loop, we do not observe a clear tendency for all nucleotides in this region to change, and the stacking and non-canonical interactions are preserved for C23, U24 and G25.

**Changes in the m6A modified system and the importance of changes in the lower bulge.**   As described above the changes observed in the lower stem loop for the unmodified system from *A* to *C* are largely similar to the changes observed going from the unmodified to the m6A-modified system (i.e. *A* to $A^*$). However, there are two alternative configurations for the bulge region (one observed in $A^*$, and the other in $B^*$ and $C^*$). The stem loop is kinking in the bulge in the *out*-configurations, while in $A^*$ there is no kink. In fact, this change in structure is the difference between the ensembles $A^*$ and $B^*$. The kinking is introduced by stacking between U9 and U12, leading to a significant change in the structure, but without the requirement to change other parts of the molecule. The new configuration will be higher in energy due to the introduced backbone strain in the bulge.

The structural changes associated with the transition to the *out*-configuration in the modified system are more modest than in the unmodified case (see Fig 6D). Two additional changes are observed in going from $A^*$ to $C^*$ in addition to the changes in the bulge discussed. The first one is located in the helical region H2, where the C18-G31 base pair is lost, changing the interactions within the helix somewhat and leading to some more stabilisation, likely due to the loss of the non-canonical interaction between m6A22 and G28 and the associated stacking. The second change is probably more interesting, as the nucleotides in the apical loop are all swinging out, similar to a fist that is transformed into an open hand, replacing the non-canonical interactions with stacking.

A final comment is reserved for the configuration of m6A22. We observe m6A22 solely in its *syn* configuration. This configuration is lower in energy, as it prevents a steric clash, but prevents WC base pairing. We observe base pairing of m6A22 through its sugar edge, and hence it can adopt this configuration in both the *in*- and the *out*-configuration.

## Discussion

The first question that needs to be answered with regard to the binding of SND1 to the ORF50 transcript is why this binding is not occurring in absence of the modification. Because no binding in absence of the modification is observed experimentally [6], and our analysis shows the

key change is the change in the *in/out*-configuration of A22, we are led to the conclusion that the *out*-configuration is likely associated with binding, and so the accessibility of this configuration is important. While the *out*-configuration exists in the unmodified stem loop, the change in free energy between the *in*- and *out*-configurations compared to the m6A-modified stem loop significantly affects its occupancy. Aside from the occupancy, it is important to consider the stability of the structures, i.e. how easy it is for the molecule to escape the funnel. The high stability of the in-configuration and the comparatively low stability of the *out*-configuration result in a very fast transition from *out* to *in*, while the opposite transition is very slow. As a result the life time of the *out*-configuration is incredibly short, and the likelihood of transitions to it incredibly small, meaning its population is approximately zero. This observation is supported by the heat capacity curves and equilibrium constants calculated from the free energy landscapes. Hence, the most likely reason no binding is observed in the unmodified case is that the required RNA configurations are simply not available.

## Changes in bulge in the lower stem are required for *out*-configurations

Given these observations, the next question is what is required for a transition to an *out*-configuration. A key observation is the absence of any *out*-configuration without a kinked bulge region, while such structures are lowest in free energy for the *in*-configuration. Likely therefore, the change in the structure lower in the stem is required to allow for changes in the configuration in and around the apical loop, where the GGACU binding motif is located. The large number of unpaired nucleotides in the bulge makes the arrangement of these nucleotides challenging, and generally at least one nucleotide sits outside the stem. The kinking in the bulge region reduces the interactions within the region and is associated with two nucleotides pointing away from the stem loop. However, this loss of interactions is associated with more flexibility and hence entropically favoured. The unkinked bulge is more stable due to the increased interactions, but as the arrangement of the nucleotides is difficult due to the crowding, it locks the stem loop structure. Hence, a change in the upper part of the stem loop is therefore linked to structural changes of the bulge region, allowing for more flexibility in the rest of the stem loop. Importantly this result matches experimental findings [6]. Further evidence for the importance of the bulge is that the absence of the bulge removes the bias in the unmethylated system. A detailed description of the energy landscape for a shortened sequence without the lower bulge is provided in S1 Text section C, with the disconnectivity for the shortened system provided in Fig C.

## m6A22 destabilises the central helix H2 and alters the bulge structure

This interaction between the apical loop and the bulge through the helical region H2 naturally leads to the question how the N6-methylation of A22 alters this behaviour. As described in Results, the methyl group requires more space, and the nucleobase is therefore moving relative to the stacked bases below. m6A22 still forms a non-canonical base pair with G28, where the changed position of the base pair alters the stacking in the helical region all the way down to the bulge. The alteration in the base pairing and stacking interactions lowers the relative stability compared to the unmodified structures in the *in*-configuration. It further primes the bulge region for the required kinking, due to changes in the helical region. These structural changes establish a connection between the modification and the changes in the lower region of the stem loop. This connection means there is a requirement for the bulge to adopt a different structure to allow for the *out*-configuration, while the modification in the apical loop affects the bulge region, leading to a connection between these mechanisms. These intertwined mechanisms provide the structural explanation of how the stem loop is destabilised upon the m6A-

modification, and how this process is linked to the lower stem loop and the RNA activation process for binding.

It should be noted at this point that these changes in H2 also affect the relative stabilities of $A^\star$ and $B^\star$. In Fig 2 this change can be seen in the representative structures shown. The orientation between H1 and H2 changes due to changes in the bulge, leading to a kink in $B^\star$, while $A^\star$ exhibits a fairly straight stem loop. These changes in the bulge require loss of the additional base pair between U12 and A38 in $A^\star$. This additional base pair and the related changes in H2 that have been described stabilise $A^\star$ compared to $B^\star$.

At this point, we can draw a comparison to the experimental findings by Baquero-Perez *et al.* [6], who provide a number of factors impacting the binding of SND1 to the ORF50 RNA transcript. Firstly, experimentally the system is metastable compared to other stem loops, and m6A-modification destabilises the system further. We observe these two features clearly in our physical modelling, namely through the existence of multiple competing structural ensembles and the changes of their relative stability as A22 is N6-methylated. Our modelling provides structural reasons for the destabilisation, and shows how the m6A-modification affects the structure of the stem loop. Furthermore, through the use of altered stem loops, it was highlighted that the lower region of the stem loop, i.e. the bulge is a key feature necessary for binding. Our model shows this link as well, and we can identify the connection of the configurations in the upper and lower parts of the stem loop through changes in stacking propagated by the helical region H2. Finally, we provide an explanation why binding is severely impaired in the unmodified stem loop.

### The opening of the apical loop allows for binding

A last comment is reserved for the changes in the apical loop upon adopting the *out*-configuration. Not only is the change exposing m6A22 to the outside, but also C23 and U24, which are both part of the described binding motif GGACU for SND1. In addition, we observe the full opening of the loop, including C25 and U26. These nucleotides form a stack that is exposed, and can likely be recognised by other molecules, including SND1. This feature leads to an appearance of the apical loop like an outstretched hand, inviting interactions.

### Conclusion

We have presented here a full investigation of the energy landscape of a 43mer stem loop of the KSHV ORF50 transcript containing a GGACU binding motif. We propose a structural explanation for the experimental observations that the m6A-modified ORF50 transcript binds to its protein partner, SND1, but does not bind if the adenine in position 22 is not N6-methylated. Our study suggests a change in the position of A22 in the methylated system with the base becoming exposed to the solvent in the modified system. This change is interconnected to other important structural modifications occurring in the loop: (a) with the bases of the apical loop also turning toward the solvent, instead of pointing inside the loop in a network of reciprocal interactions, and (b) alterations in the helix (H2) close to the modified nucleotide, which extend all the way to the central bulge separating the two helical regions of the stem-loop. These conclusions from our modelling analysis are in agreement with the experimental evidence suggesting that structural changes have to occur between the methylated and the native system for the binding to occur. In particular, our suggestion that a restructuring of the whole apical portion of the stem-loop (H2 and loop) is needed to accommodate for the methylated nucleotide is in agreement with the observation that the stem-loop is destabilised and is rearranged in the methylated system.

In our analysis we are able to pinpoint the behaviour of key nucleotides undergoing significant structural changes between the two systems. These structural details open up the possibility of performing new experiments targeting specifically these fine structural details (such as chemical probing). If confirmed, our hypothesis would give a full structural picture of the two systems, the relationship between their structures, and their ability to bind. Moreover, the structure proposed for the modified system could be used for further modelling and experimental studies of the ORF50 system in the presence of the protein, or at least of the portion of the protein known to be involved in binding.

On a more general note, we apply here a new method to study the details of chemically modified RNA structures with a focus on the ensembles of possible alternative structures that the system might adopt. With systems as flexible and polymorphic as RNAs, where multiple structures are frequently observed for a given sequence, this approach is key to be able to correctly interpret experimental data and to link structures with experiments, gaining a coherent view of the system's behaviour. We had applied this approach successfully in the past on a system for which several alternative experimental structures were resolved and for which mutational data was available, and in this work we report the first example of a study of the changes of the structural ensembles upon epigenetic modifications without any available experimental three-dimensional structures.

## Supporting information

**S1 Text. Additional information for the methodology, energy landscape for a shortened loop, and more detail on the CV curve. Section A**: Additional detail for the methodology. A comment on why the exploration of energy landscapes is desirable, and how it is practically achieved. Details on how to judge sampling convergence in general are also included, and how these points apply to the present study. **Section B**: Additional structural information and more detailed structural representations. **Section C**: Description of the energy landscape for a shortened sequence, including the used methodology, a brief discussion of the results, and a disconnectivity graph of the free energy landscape. **Section D**: Additional analysis of the CV curves. Description of the third peak in the CV curves. **Figure A**: Initial structure obtained from RNA-composer. The two-dimensional structure from experiment and the corresponding three-dimensional structure obtained from it. **Figure B**: Structural close-ups. Comparison of the apical loop, the upper helix H2 and the bulge for $A$, $C$, $A^\star$ and $C^\star$. **Figure C**: Free energy disconnectivity graph at 310 K for the shortened unmethylated loop. **Figure D**: Example structures for the high temperature transition.
(PDF)

## Author Contributions

**Conceptualization:** Konstantin Röder, Amy M. Barker, Adrian Whitehouse, Samuela Pasquali.

**Data curation:** Konstantin Röder.

**Formal analysis:** Konstantin Röder, Samuela Pasquali.

**Funding acquisition:** Adrian Whitehouse.

**Investigation:** Konstantin Röder, Samuela Pasquali.

**Methodology:** Amy M. Barker, Adrian Whitehouse, Samuela Pasquali.

**Resources:** Konstantin Röder.

**Validation:** Samuela Pasquali.

**Visualization:** Konstantin Röder, Samuela Pasquali.

**Writing – original draft:** Konstantin Röder, Samuela Pasquali.

**Writing – review & editing:** Konstantin Röder, Amy M. Barker, Adrian Whitehouse, Samuela Pasquali.

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
