## [Decision Letter · Decision Letter 0]

14 Jan 2022

Dear Dr Röder,

Thank you very much for submitting your manuscript "Investigating the structural changes due to adenosine methylation of the  Kaposi’s sarcoma-associated herpes virus ORF50 transcript" (PCOMPBIOL-D-21-02101) for consideration at PLOS Computational Biology. As with all papers peer reviewed by the journal, your manuscript was reviewed by members of the editorial board and by several independent peer reviewers. Based on the reports, we regret to inform you that we will not be pursuing this manuscript for publication at PLOS Computational Biology.

We greatly appreciate the significant importance of the research subject and the conclusions derived from the study. However, all the reviewers raised serious concerns about the validity of the main conclusions, and none of them recommended the publication of the manuscript. Particularly, reviewers questioned the potential bias of the result caused by the specific choice of the initial structures.

The reviews are attached below this email, and we hope you will find them helpful if you decide to revise the manuscript for submission elsewhere. We are sorry that we cannot be more positive on this occasion. We very much appreciate your wish to present your work in one of PLOS's Open Access publications. 

Thank you for your support, and we hope that you will consider PLOS Computational Biology for other submissions in the future.

Sincerely,

Shi-Jie Chen

Associate Editor

PLOS Computational Biology

Arne Elofsson

Deputy Editor

PLOS Computational Biology

Reviewer's Responses to Questions

**Comments to the Authors: **

Reviewer #1: Roder et al. investigate a stem-loop within ORF50 of KSHV by exploring the energy landscapes of the loop unmethylated and methylated at A22. They detail the structural differences between the unmethylated and methylated as well as between the in-configuration and out-configuration. The manuscript is well-written, and the science appears sound. My main concerns are as follows:

1. As I was reading the manuscript, I was hoping to see the sequence and a general secondary structure representation of the stem-loop being discussed. Looking at Fig. 6 was helpful, but this seemed way too late in the manuscript. I realize that the secondary structure between the unmethylated and methylated can be different as well as the secondary structure between the in- and out-configuration. But, having some visual to set the stage early would be helpful.

2. Three sets of BH runs were conducted. For one set, an unfolded structure was used as the starting structure. How did the structures resulting form the RNAComposer starting structure compare to the structures resulting from the unfolded structure? Would there be any benefit to starting with another unfolded (or random) structure?

3. I was hoping to see more detailed structures showing the atom-atom interactions. For example, line 133 states that A22 “interacts with the surrounding nucleotides.” Is it worth showing these interactions in a figure? Similarly, lines 235-239 discuss stacking in the unmodified system compared to the modified system and a different configuration of A22. Perhaps a detailed figure would help here too.

4. The authors conclude that the changes in the lower region and upper loop are connected (see “Changes in the bulge in the lower stem are required for out-configurations” section). What would happen if a smaller system was explored that consisted only of the upper loop (no bulge in the lower stem). If the conclusion is valid, no out-configurations would result. Can this be done to add validity to the conclusion?

5. While reading the “The in-configuration in the unmodified and m6A-modified stem loop” section, it wasn’t always obvious to me when the authors were referring to the unmodified versus the modified loop. Perhaps they could add some distinction there.

6. In Figs. 1 and 2, perhaps the authors could mention that A22 is colored red in the structures.

7. I was a little confused by the drawings in Figs. 3 and 4. In Fig. 4, I understand the two structures in equilibrium as represented by P1. I do not understand the equilibria where one structure is drawn on one side of the arrow and two structures were drawn on the other side. Is the one structure in equilibrium with both structures?

8. The authors didn’t mention the shoulder in Fig. 3 and the additional peak in Fig. 4. Any ideas what they represent?

9. In comparing Fig. 3 to Fig. 4, I did notice that the transition energy of P1 in Fig. 3 is very similar to the transition energy of P2 in Fig. 4. The same is true for the transition energy of P2 in Fig. 3 to the unlabeled peak in Fig. 4. Is there any significance to this? Similarly, in Fig. 4, why isn’t the first peak unlabeled, the second peak P1, and the third peak P2 (making the transition energies of P1 and P2 very similar in both Figs. 3 and 4)?

10. I was a little confused by Fig. 5. What do the pink and orange strips represent? I don’t understand the y-axis for the first three panels (i.e. what are 1, 0.5, and 0 in relation to canonical base pairing?). I was also confused by the fact that, in the top panel for example, why there is a 1 on the top half of the plot and a 1 on the bottom half of the plot. Similarly, in the bottom plot, there are four C3’-endo labels on the y-axis. Why?

11. In Fig. 6, can the blue and red frames be made darker and thicker? I also understood the “in-configuration” and “out-configuration” to refer to A22. In the caption for panel b, the authors use these terms to describe the lower bulge, which was confusing.

Minor corrections:

1. Line 11 – “Stimulated” is misspelled.

2. Line 16 – “Transcriptome” is misspelled.

3. Line 26 – “recruitment the” should be “recruitment of the.”

4. Line 32 – I am not familiar with DRACH. Is it an acronym?

5. Line 161 – “maybe” should be “may be.”

6. Line 183 – “We can associated” should be “We can associate.”

7. Line 205 – “Other key region” should be “other key regions.”

8. Line 227 – “non-canonical interaction” should be “non-canonical interactions.”

9. Fig. 5 – the “stacking” label is misspelled.

Reviewer #2: This manuscript titled “Investigating the structural changes due to adenosine methylation of the Kaposi’s sarcoma-associated herpesvirus ORF50 transcript” describes structural rearrangement in 43 mer stem-loop RNA due to m6A post-transcriptional chemical modification in its one of the bases. Authors have explored in and out configurations of A22 nucleotide in unmodified and 22m6A modified RNA structures. Authors have also revealed structural changes in stem-loops using the energy landscape framework. In order to obtain low energy structures author has used well known basin-hopping techniques and performed sufficient runs. Authors have also studied the kinetic transition network and its transition states using discrete path sampling and DNEB algorithm as well as hybrid eigenvector. Authors have studied rate constants and equilibrium constants of in and out the configuration of the unmodified and modified system, where they observed a significant number of out configurations in case of modified (m6A22) system. Authors have well explained that how m6A modification of A22 base would result in exposed nucleotide outwards that would be key for the highly specific interaction with reader protein such as SND1. This manuscript is well written, and mages were well organized and properly labelled. However, I have the following major and minor concerns:

1) The WT and m6 modified RNA stem-loop structures studied here using computational methods are modelled structures. Are there any experimentally determined structure (3D or even 2D probing based structure) available to make sure that we are starting with the right structure? My worry is that depending on the starting structure (for example different arrangement of initial base pairs in the structure) we may get a different result. On what basis is the starting structure of WT and m6A modified RNA different? Is there any experimental evidence for this available in the literature for the starting structures? Is there any experimental evidence as to which starting structure is more stable: WT vs m6A mutant?

2) Authors have used a previous study that reported that tudor SND1 protein is an m6A RNA reader essential for Kaposi sarcoma-associated herpesvirus as the justification for explaining the results here. That study however looked at m6A methylation and the role of SND1 as an m6A reader on a global scale using high throughput sequencing method. This study looked at things at a global scale on full-length RNA. The right study to compare the results would be when SND1 and the short stem-loop RNA (WT and m6A modified) would be used, showing the differences in binding. Is there any biochemical/biophysical evidence that showed that Tudor SND1 binds to the m6A modified stem-loop and not WT stem-loop RNA?

3) In a result, authors related structural changes with heat capacity curves modified and unmodified systems and well-explained energy scale with Boltzmann population proportions for P2 peak of the modified system and observed a higher percentage of molecules compared to unmodified system. In figure 4, there is a third peak on the right-hand side of P2 peak, which is at a higher transition energy state. Can the authors explain this third peak?

Minor comments

4) Page 2 (introduction): Is it “KSHV” or “KHSV” in the sentence, “Several studies have demonstrated the KHSV transcriptome is heavily m6A methylated.”?

5) Page 4 (last paragraph): the authors discussed the appearance of more distinct sub-funnels within in-configuration in the modified system due to changes in the lower stem-loop. However, there are alterations also present in the unmodified system but with no separate distinction of sub-funnels. 

6) Table 1, Figure 1, 3, and 4: Is it “A23” or “A22”? It should be A22.

7) Page 5 (last paragraph line 183): “associated” can be “associate.” 

8) Page 6 (last paragraph line 224): The pairing is non-canonical (GU base pair) instead of canonical.

Reviewer #3: The authors present a theoretical work addressing the effect of an m6A modification on the structural ensemble associated to a RNA stem loop. This modification is known to affect binding with a m6A reader (SND1). In this work, a significant effect of the methylation on the population of different states is observed. This effect is suggested to be responsible for the increased affinity of the methylated motif with SND1. The work is difficult to understand and, in my opinion, results are not correctly interpreted. As such, I think that publication is premature at this stage.

My main concern regards the results presented in Figures 1 and 2. The results are difficult to rationalize. Whereas I understand that the structure of the apical loop should be affected by the methylation, I don't understand how the structure of the lower part of the system could be correlated with the structure of the loop. This seems to be a key issue, since it is an important difference between the ensembles represented in Figure 1 and 2. The authors should provide an explanation. My suspect is that this is just a consequence of the random initialization of structures in the modified and non modified simulations.

The same problem emerges in Figure 6: how is the methyl group affecting the structure of the lower bulge (panel a)? There's no explanation for this, and I guess this is a random result.

A much more robust result could be obtained by using the same initial structures for the modified and not modified ensembles, just minimizing them separately with the modified / unmodified force fields. Are the authors doing this? As far as I understood, with the adopted procedure, it is extremely likely that randomness in the construction of the ensembles dominate the result.

Line 74-80: "Three sets [...] landscapes." I am not sure I understand what the authors did. Are the unfolded structures (third set) modified or not? Isn't this choice leading to a different number of initial modified vs unmodified models? What's the rationale of this choice? Are the authors just building a large database and picking the 100 lowest energy structures? How many initial structures (generated with RNA composer) were used?

Line 118-120: Taken literally, the authors are claiming that the methyl group induces an energy shift of 12 kcal/mol. There is no explanation for such a large difference. 

Line 133-138 and 293: I think there is a logical flaw here. The authors write the text as if the rates were a consequence of the free energy differences between the local minima. This is not correct: the rates are indeed a consequence of the differences between the local minima and the transition states. The rate between forward and backward transition rates then is the reason for the observed population. For instance, at line 293 the authors write that "the high stability of the in configuration [...] result in a very fast transition rate". This is logically incorrect.

Line 290: According to the prediction of the authors, the out configuration does NOT exist in practice in the unmodified stem loop (22-30 kcal/mol implies a negligible population).

Table 1: I cannot understand how a difference of 22-30 kcal/mol (see line 118) or of 10 kcal/mol (see line 120) can result in the equilibrium constants reported in the table.

Line 158 "A useful way". I cannot see how this representation can be useful. As far as I understand, this heat capacity is not related to anything that can be measured experimentally.

Figure 5: there's no explanation for the labels in the last panel. Why is C3'-endo repeated?

Lines 364-374: As far as I understand from this text, the experiment suggests that there is a conformational change driven by methylation. It does not suggest that the lower bulge should be affected by the methylation. So, this finding, which is puzzling (see above), is not validated in any way.

Minor issues:

Line 11 Typo ("Stimualted")

Line 98: I would add a reference rather than showing the AMBER keyword (igb=2)

Line 201: "were" -> "where"

In several places, the authors write A23 instead of A22 (can be found with text search).

Figure 5: staking -> stacking

Figure 5, caption: "Barbnaba" -> "Barnaba"

**Have the authors made all data and (if applicable) computational code underlying the findings in their manuscript fully available?**

Reviewer #1: Yes

Reviewer #2: Yes

Reviewer #3: Yes

PLOS authors have the option to publish the peer review history of their article (what does this mean?). If published, this will include your full peer review and any attached files.

Reviewer #1: No

Reviewer #2: No

Reviewer #3: No

---

## [Decision Letter · Decision Letter 1]

12 Apr 2022

Dear Dr Röder,

Thank you very much for submitting your manuscript "Investigating the structural changes due to adenosine methylation of the  Kaposi’s sarcoma-associated herpes virus ORF50 transcript" for consideration at PLOS Computational Biology. As with all papers reviewed by the journal, your manuscript was reviewed by members of the editorial board and by several independent reviewers. The reviewers appreciated the attention to an important topic. Based on the reviews, we are likely to accept this manuscript for publication, providing that you modify the manuscript according to the review recommendations.

Sincerely,

Shi-Jie Chen

Associate Editor

PLOS Computational Biology

Arne Elofsson

Deputy Editor

PLOS Computational Biology

[LINK]

Reviewer's Responses to Questions

**Comments to the Authors:**

Reviewer #1: Roder et al. investigate a stem-loop within ORF50 of KSHV by exploring the energy landscapes of the loop unmethylated and methylated at A22. They detail the structural differences between the unmethylated and methylated as well as between the in-configuration and out-configuration. The authors adequately addressed my concerns related to the original submission. Upon reading the revised manuscript, I have a few minor concerns:

On line 33, when the authors use a footnote to define DRACH, perhaps they could include that R = A or G.

On line 70, “information contain further detail” should be “information contains further detail.”

On lines 75 and 211 and in SI section S2 (three times) “figure” should be capitalized.

On line 149, “orders of magnitudes slower” should be “orders of magnitude slower.”

On line 266, “alternation leads changes” needs to be corrected.

On line 290, “asides from the changes” needs to be corrected.

On line 337 when the authors refer to the SI, perhaps they could direct the reader to section S3.

On line 342, “described in the results section” can be “described in Results.”

On line 368, “explanation while binding” needs to be corrected.

Figs. S3 and S4 aren’t mentioned in the main text. See below for more about Fig. S4.

In the caption to Figs. 3 and 4, the authors should refer the reader to SI section S4 and Fig. S4 for further information about the shoulder (Fig. 3) and additional peak (Fig. 4).

In the SI, “BARRIER find lower” should be “BARRIER finds lower.”

In the SI, “shorten stem loop” should be “shortened stem loop.”

In the caption to Fig. S3, “disconnectivity” is spelled incorrectly.

Reviewer #2: I had raised a few major concerns in the first review of this paper. One of the concern I had asked for the basis on which the selection of starting RNA structure was made. Authors have explicitly explained that the experimental 2D structure of the RNA was used to generate the 3D model of RNA using RNAcomposer. The results suggest that WT in-configuration is more stable. In my second major concern, I had asked if there is any experimental evidence for binding of tudor SND1 to m6A modified stem-loop RNA and not with the WT stem-loop RNA. Authors have explained this point by citing the published experimental results. In an another question, I had asked for an explanation for the occurrence of third peak at higher transition energy state in figure 4. Authors have explained that this third peak corresponds to the loss of structural features, such as stacking and base pairing, and is associated with a transition form folded to partially folded structures

Therefore, overall authors have satisfactorily explained my concerns.

Reviewer #3: The authors replied to my questions. It turned out that most of my doubts were related to unclear explanations. To be fair, I still do not understand some of the presented results and I would encourage the authors to clarify them for the benefit of the readers:

1. When comparing Fig 1 and Fig 2, I see two major effects: m6A stabilizes C wrt B and m6A stabilizes A wrt B. The former makes sense (I still do not understand how the change can be this large, but I trust the authors). The second I suspect is due to the randomness of the basin hopping algorithm. If this is correct, please comment this in the caption.

2. If I understand correctly, the discrepancy between the large free energy differences and the relatively small Keq (Table 1) is a missing entropic effect associated to configuration count. If this is correct, please explain this in more detail in the paper. Readers will implicitly associate free energies with populations and macrostates, whereas here the authors are reporting free energies for micro states, and the number of these states is different in different macro states. Please also report an estimate in the count of the number of states that could justify this discrepancy.

3. Related to the previous point, is the horizontal density of points in Figs 1 and 2 uniform? If so, can I deduce that basin hopping is NOT able to report the correct number of states that would be required to correctly compute populations of macro states? Again, a comment on this issue would help the reader.

Finally, one small issue that I didn't mention in my first review is that it might be interesting to know if m6A is in syn or anti conformation in the in- and out- state. In general, m6A is expected to be anti when WC paired and syn when non WC paired (see e.g. https://pubmed.ncbi.nlm.nih.gov/25611135/). Here, in the in-state it is forming a non canonical pair with G28 (Fig. 6c). It would be interesting to know if this pairing requires A22 to flip to the least stable anti conformation or not. The analysis should be straightforward and the result could be useful.

**Have the authors made all data and (if applicable) computational code underlying the findings in their manuscript fully available?**

Reviewer #1: Yes

Reviewer #2: Yes

Reviewer #3: Yes

PLOS authors have the option to publish the peer review history of their article (what does this mean?). If published, this will include your full peer review and any attached files.

Reviewer #1: No

Reviewer #2: No

Reviewer #3: No

Figure Files:

Data Requirements:

Reproducibility:

References:

---

## [Editor Report · Decision Letter 2]

28 Apr 2022

Dear Dr Röder,

We are pleased to inform you that your manuscript 'Investigating the structural changes due to adenosine methylation of the  Kaposi’s sarcoma-associated herpes virus ORF50 transcript' has been provisionally accepted for publication in PLOS Computational Biology.

Best regards,

Shi-Jie Chen

Associate Editor

PLOS Computational Biology

Arne Elofsson

Deputy Editor

PLOS Computational Biology

---

## [Editor Report · Acceptance letter]

18 May 2022

PCOMPBIOL-D-21-02101R2 

Investigating the structural changes due to adenosine methylation of the  Kaposi’s sarcoma-associated herpes virus ORF50 transcript

Dear Dr Röder,

I am pleased to inform you that your manuscript has been formally accepted for publication in PLOS Computational Biology. Your manuscript is now with our production department and you will be notified of the publication date in due course.

With kind regards,

Livia Horvath
